# Validation of a Commercial Loop-Mediated Isothermal Amplification (LAMP) Assay for the Rapid Detection of *Anisakis* spp. DNA in Processed Fish Products

**DOI:** 10.3390/foods9010092

**Published:** 2020-01-16

**Authors:** Gaetano Cammilleri, Vincenzo Ferrantelli, Andrea Pulvirenti, Chiara Drago, Giuseppe Stampone, Gema Del Rocio Quintero Macias, Sandro Drago, Giuseppe Arcoleo, Antonella Costa, Francesco Geraci, Calogero Di Bella

**Affiliations:** 1Istituto Zooprofilattico Sperimentale della Sicilia, via Gino Marinuzzi 3, 90129 Palermo, Italy; vincenzo.ferrantelli@izssicilia.it (V.F.); antonella.costa@izssicilia.it (A.C.); francesco.geraci@izssicilia.it (F.G.); calogero.dibella@izssicilia.it (C.D.B.); 2Dipartimento di Scienze della Vita, Università degli studi di Modena e Reggio Emilia, Via Università 4, 41121 Modena, Italy; andrea.pulvirenti@unimore.it; 3Enbiotech s.r.l. Via Aquileia 34, 90144 Palermo, Italy; c.drago@enbiotech.eu (C.D.); g.stampone@enbiotech.eu (G.S.); g.delrocioquintero@enbiotech.eu (G.D.R.Q.M.); s.drago@enbiotech.eu (S.D.); g.arcoleo@enbiotech.eu (G.A.)

**Keywords:** *Anisakis* spp., molecular methods, LAMP, validation, anisakidae family

## Abstract

Parasites belonging to the *Anisakis* genera are organisms of interest for human health because they are responsible for the Anisakiasis zoonosis, caused by the ingestion of raw or undercooked fish. Furthermore, several authors have reported this parasite to be a relevant inducer of acute or chronic allergic diseases. In this work, a rapid commercial system based on Loop-Mediated Isothermal Amplification (LAMP) was optimised and validated for the sensitive and rapid detection of *Anisakis* spp. DNA in processed fish products. The specificity and sensitivity of the LAMP assay for processed fish samples experimentally infected with *Anisakis* spp. larvae and DNA were determined. The LAMP system proposed in this study was able to give positive amplification for all the processed fish samples artificially contaminated with *Anisakis* spp., giving sensitivity values equal to 100%. Specificity tests provided no amplification for the *Contracaecum*, *Pseudoterranova*, or *Hysterothylacium* genera and uninfected samples. The limit of detection (LOD) of the LAMP assay proposed was 10^2^ times lower than the real-time PCR method compared. To the best of our knowledge, this is the first report regarding the application of the LAMP assay for the detection of *Anisakis* spp. in processed fish products. The results obtained indicate that the LAMP assay validated in this work could be a reliable, easy-to-use, and convenient tool for the rapid detection of *Anisakis* DNA in fish product inspection.

## 1. Introduction

The Anisakidae family includes a vast number of parasites with a worldwide distribution. The life-cycle of Anisakidae nematodes involves invertebrates, fish, cephalopods, and marine mammals so these parasites can be found in the muscles and viscera of numerous fish and cephalopod species [1,2,3,4,5]. In the Mediterranean, the *Anisakis* spp. parasites can be found in different teleosts belonging to different ecological distributions [6]. The fish species mainly involved in the life cycle of *Anisakis* belong to the pelagic, benthopelagic, and benthodemersal domains. Indeed, the three *Anisakis* species are widely occurring in pelagic, benthopelagic, and demersal species of the Gadidae, Merlucciidae, Scombridae, Carangidae, and Trichiuridae families [2,7,8]. *Anisakis* spp. parasites are marine nematodes of health interest because of their high zoonotic potential, being responsible for a human disease called Anisakiasis. Anisakiasis is a zoonotic disease caused by the ingestion of raw or undercooked fish infected with Anisakidae larvae [9,10,11,12,13]. Anisakiasis has become an increasing human health concern, particularly in Asian countries, representing more than 50% of all cases, where the consumption of raw or undercooked fish is frequent and/or has become increasingly popular. The remaining cases are from European countries, especially Italy (57 cases from 1996 until now), Spain (124 cases from 1996 until now), and France (15 cases from 1975 until now), whereas there are much fewer cases in Scandinavian countries, despite comparatively high fish consumption rates per capita in these countries [7,14]. Moreover, several authors have reported this parasite to be a relevant inducer of acute or chronic allergic diseases [15,16,17,18]. *Anisakis* is implicated in allergic IgE-mediated reactions, such as urticaria, angioedema, asthma, and anaphylaxis, in highly sensitized people [19,20]. The European Food Safety Authority (EFSA 2010) confirmed that all wild saltwater fish must be considered at risk of containing viable parasites of human health concern and no sea fishing grounds can be considered free of *A. simplex complex* larvae [21].

EFSA also recommends further studies and methods to improve the surveillance and diagnostic awareness of pathologies to parasites in fishery products. Even the Italian Ministry of Health encourages fish sector operators to carry out correct evisceration protocols of fish products in order to prevent *Anisakis*-related pathologies by reducing the possibility of migration of L3 larvae in the musculature [22]. At present, European authorities perform laborious and unreliable inspection methods, such as visual control and transillumination with UV [23], which cannot be applied for processed fish products such as anchovy paste, marinated anchovies, infant formulas, etc. [24]. Furthermore, the current immunological methods for *Anisakis* allergy diagnosis give a high number of false positives due to the cross-reactivity with numerous panallergens [25,26].

In this case, molecular biology methods are valuable tools in the detection of *Anisakis* spp. nematodes in processed seafood [24,27,28,29,30]. Several studies have shown that immunological and molecular methodologies yield comparable results concerning the detection of allergens in processed foods as sensitive and specific tools [31,32,33]. According to literature, Real-Time PCR is recognized to be the only molecular method capable of detecting the presence of *Anisakis* spp. in processed fish products, showing satisfactory sensitivity and specificity values [24,27,30,33,34]. Nonetheless, this sophisticated and expensive molecular method is currently indispensable and requires operational skills limiting their broad applicability. Loop-mediated isothermal amplification (LAMP) is considered a highly sensitive and rapid method for DNA amplification at constant temperature (60–65 °C) [35,36].

The food sector operators must have systems and procedures that allow competent authorities to access information on the products in order to guarantee their hygiene. In this work, a commercial LAMP assay for *Anisakis* spp. DNA detection was optimised and validated in order to obtain a simple, fast, and cheap tool, which can identify possible risks to consumer health due to the presence of these organisms in processed fish products.

## 2. Materials and Methods

### 2.1. Fish Samples and *Anisakis* Larvae Collection

All the processed fish samples used for the method optimisation and validation came from a large-scale distribution, in order to reduce any bias from local food specialities and extend the range of validation. Homogenised farmed trout (*Oncorhynchus mykiss*; *n* = 40), homogenised farmed sea bream (*Saprus aurata*; *n* = 40), and homogenised farmed salmon (*Salmo salar*; *n* = 40) were chosen as naturally negative (noncontaminated by *Anisakis*) samples [4,21,37,38,39]; whereas anchovy (*Engraulis encreasicolus*) paste (*n* = 40), anchovy in oil (*n* = 40), and salted sardines (*Sardina pilchardus*; *n* = 40) samples were chosen as positive samples for the validation of the method and for matrix effects evaluation. All the processed fish samples came from Italian supermarkets. The Anisakidae larvae used for the artificial contamination of the samples were collected from *Lepidopus caudatus*, *Clupea harengus*, and *Merluccius merluccius* samples after visual inspection and modified chloro-peptic digestion [40]. The larvae isolated were washed in physiological saline serum (pH 7) and morphologically identified by B-800 light microscopy (Optika, Ponteranica, Italy) according to taxonomic keys [41]. Furthermore, *Anisakis* morphotype II, *Hysterothylacium* sp., *Contracaecum* sp., and *Pseudoterranova* sp. larvae were taken from the reference materials of the Centro di Referenza Nazionale per le Anisakiasi (Palermo, Italy). The number of *Anisakis* spp. larvae used for artificial contamination has been defined according to the prevalence of infestation of the fish species examined and reported in the literature [8,42,43,44]. The larvae were cut into small pieces and then were carefully mixed with the processed fish samples.

### 2.2. DNA Extraction

Genomic DNA was extracted from positive and negative fish samples, contaminated or not with *Anisakis* spp., respectively. The extraction was also carried out for the samples artificially infested with *Contracaecum* sp., *Pseudoterranova* sp., and *Hysterothylacium* sp. The DNA extraction was performed using a ready-to-use buffer contained in the Anisakis Screen Glow kit (Enbiotech S.r.l., Palermo, Italy). Then, 250 ± 50 mg of sample was directly placed into 15 mL tubes containing 4 mL of the ready-to-use extraction buffer (Enbiotech S.r.l., Palermo, Italy) and then incubated for 40 ± 5 min at room temperature.

### 2.3. Primers Design and LAMP Assays

To design the primer set targeting *Anisakis* spp. gene, the genomic sequences of the internal transcribed spacer 2 gene from various species were collected from GenBank^TM^ (EU624342.1, AY826720.1, AB277823.1, AB196671.1, AB277821.1, AB551660.1, HF911524.1, AY826723.1, EU718479.1, JQ912690.1, KF512840.1, JX535521.1, KF032062.1, EU327691.1). A set of six primers, two outer (F3 and B3), two inner (FIP and BIP), and two loop (LF and LB), which recognised eight distinct regions of the target gene, was designed.

The analytical and diagnostic assays to recognise *Anisakis* spp. DNA was performed using Anisakis Screen Glow commercial kit (Enbiotech Group S.r.l., Palermo, Italy) with ICGENE mini portable instrument (Figure 1) (Enbiotech Group s.r.l., Palermo, Italy), consisting of a real-time fluorimeter, monitored and regulated by the the ICGENE application (Enbiotech Group s.r.l., Palermo, Italy), downloadable on various smart devices. The Anisakis Screen Glow commercial kit includes ready-to-use reaction tubes (containing primers, fluorescent dye, etc.) to achieve a rapid amplification of DNA template. The protocol to obtain the specific amplification of the target *Anisakis* spp. DNA was carried out in a mixture of a final volume of 55 μL, including 22 μL of Anisakis Screen Glow LAMP mix (Enbiotech Group s.r.l., Palermo, Italy), 30 μL of mineral oil, and 3 μL of the extracted DNA samples. The mineral oil was added to the top of the reaction mixture to prevent evaporation. The amplification was optimised and performed at 65 °C for 35 min. Real-time monitoring of the fluorescence associated with the amplification was possible using the fluorimeter of the ICGENE portable instrument and the ICGENE application interface.

### 2.4. Specificity of the LAMP Assay

Based on the evolutionary relationships and their feasible genetic similarity, parasitic material belonging to the Anisakidae and Raphidascaridadae family was screened by the method proposed to have evidence on the diagnostic specificity of the LAMP assay. The parasitic material was previously characterised as belonging to *Hysterothylacium*, *Contracaecum*, and *Pseudoterranova* genera using specific molecular diagnostic keys and methods reported in the literature [45,46,47]. Two different experiments were carried out for the specificity evaluation: (i) In the first experiment performed using genomic DNA, *P. decipiens* sensu stricto, *P. krabbei*, *P. cattani*, *P. azarasi*, *C. rudolphii* A, and *Hysterothylacium aduncum* were tested in duplicate, also including internal positive and negative controls of the Anisakis Screen Glow commercial kit (Enbiotech S.r.l., Palermo, Italy); (ii) in the second experiment, processed fish samples artificially contaminated with *Pseudoterranova* sp., *Contracaecum* sp., and *Hysterothylacium* sp. larvae were subjected to DNA extraction and analysed in duplicate, together with positive and negative internal controls. Twenty samples of each processed fish product type, of which ten were artificially infested (5 with genomic DNA and 5 with larvae), were tested for the specificity assessment.

### 2.5. Sensitivity of the LAMP Assay

The sensitivity or inclusivity [48] was established as the ability of the LAMP method to detect DNA of *Anisakis* spp. (expressed as a percentage). Two different experiments were carried out for the inclusivity evaluation: (i) In the first experiment performed using genomic DNA, *A. pegreffii*, *A. simplex* sensu stricto, *A. typica*, *A.ziphidarum*, and *A. physeteris* were tested in duplicate, also including internal positive and negative controls; (ii) in the second experiment, processed fish samples artificially contaminated with *Anisakis* spp. type I and type II larvae were subjected to DNA extraction and analysed in duplicate, together with positive and negative internal controls. Twenty samples of each processed fish product type were tested for the inclusivity assessment.

### 2.6. Limit of Detection

The limit of detection (LOD) was established as the lowest concentration of DNA of *Anisakis* species, which provided a significantly different signal to the negative control. The determination of the LOD of the LAMP method was assessed by serial 10-fold dilution of the DNA extracted from *Anisakis* spp. larvae with nuclease-free water. All measurements were performed in ten replicates from each sample type independently. The range of the DNA extracted varied between 2.22 ng µL^−1^ and 8.40 ng µL^−1^ with good 260/280 and 260/230 ratios (1.8 to 2.1). A test was considered acceptable when it ensured the detection of positive samples successfully with DNA content equal to or greater than LOD.

### 2.7. Real-Time PCR Assay

Real-Time PCR (RT-PCR) assays of the same samples analysed by LAMP were also carried out for a comparative purpose and as a confirmation method for the results of the LAMP assays. The RT-PCR amplification was carried out using the materials and following the validated protocols described by Cavallero et al. (2017) [27]. The extraction of DNA from artificially or noncontaminated processed fish samples was carried out using the Ion Force Fast kit (Generon, Modena, Italy), following the manufacturer’s instructions. The PCR reactions were carried out by the commercial kit PATHfinder Anisakis/Pseudoterranova DNA detection assay (Generon, Modena, Italy). Five microliters of DNA extracted were mixed with 15 µL of PATHfinder Anisakis/Pseudoterranova kit for a total volume of 20 µL. The RT-PCR was carried out in a BIOER 9600 series Thermocycler (BIOER, Hangzhou, China) following the present thermal cycling conditions: A Taq activation at 95 °C for 3 min, and 45 cycles of amplification (95 °C for 10 s and 57 °C for 16 s).

## 3. Results

### 3.1. Optimization of the LAMP Assay

The method was optimised for DNA extraction phase by testing in triplicate the initial weight of the samples at 50, 100, 250, and 350 mg with 0.5, 2, 4, and 8 mL of extraction buffer. The extract was tested with undiluted and diluted 1:5 and 1:10 for matrix effect assessment. An initial weight of 250 mg with 4 mL of extraction buffer and a 1:5 dilution after the extraction were found to be the best conditions for effective real-time detection of DNA amplification by the LAMP method proposed, giving the fluorescence intensity required for detection. The method was optimised by real-time monitoring of the time and temperature of reaction. The optimal reaction temperature and time for the LAMP assay was proved to be 65 °C and 35 min, respectively.

### 3.2. Sensitivity and Specificity

The LAMP method proposed was able to amplify *Anisakis* spp. DNA from artificially infested fish samples, giving a sensitivity of 100% for each sample type analysed (Figure 2). Moreover, the method was also able to detect each sample contaminated with *A. simplex* s.s., *A. pegreffii*, *A. physeteris*, *A. ziphidarum*, and *A. typica* DNA. The assay detected *Anisakis spp.* DNA to a dilution of 10^−4^ (0.00022 ng µL^−1^), giving an amplification for all the replicates, with fluorescence values necessary for detection.

All the LAMP analysis were carried out using positive and negative controls contained in the Anisakis Screen Glow kit. No amplification products were detected in uninfected samples, giving a very high specificity rate. Furthermore, no amplification was obtained on processed fish samples contaminated with *Contracaecum* sp., *Pseudoterranova* sp., and *Hysterothylacium* sp. larvae and DNA. The RT-PCR assay showed a sensitivity rate of 100% for the fish samples artificially infested with *Anisakis* sp. larvae and DNA. However, the analysis of uninfected samples and samples contaminated with *Pseudoterranova* sp., *Hysterothylacium* sp., and *Contracaecum* sp. larvae and DNA for the specificity test did not obtain any amplification.

In agarose gel analysis, the LAMP amplicons revealed a ladder-like pattern, according to what was reported in the literature, with many bands of different molecular weights, indicating the production of stem–loop DNA with inverted repeats of the target sequence (Parida et al., 2008; Li et al., 2012; Figure 3).

## 4. Discussion

The method optimised with the set of primers employed was able to amplify *Anisakis* spp. DNA in 35 min at 65 °C from a considerable initial weight of samples, giving a satisfactory sensibility and specificity. The time needed for the LAMP assay was much lower than the RT-PCR method (about 94 min). The use of two loop primers and the ability of the two sets of primers (F3–B3, FIP–BIP) to recognise two distinct regions on the target DNA allowed acceleration of the reaction time and provided extremely high specificity. The specificity test has demonstrated that the LAMP assay proposed in this work can discriminate the *Anisakis* genera with respect to species belonging to other related genera that are not present in the Mediterranean Sea, such as *Pseudoterranova* spp. Furthermore, the method was able to discriminate the *Anisakis* genera with respect to nonallergic parasites such as *Hysterothylacium* spp, which proved that the LAMP primers are highly specific for the detection of *Anisakis* spp. The LAMP reaction primers recognise specifically eight independent regions in comparison with PCR primers, which can only recognise two independent regions, enhancing the sensitivity and specificity and decreasing the probability of false-positive results [49]. Generally, the fluorescence-based real-time monitoring of LAMP reaction is considerably faster than that performed by a real-time turbidimeter [50]. Moreover, compared to the real-time turbidity method, the real-time fluorescence method possesses two further advantages: The first is higher sensitivity; the second is that the sensitivity is less affected by the presence of opaque substances in the mixture, such as proteins [51]. The LAMP assay optimised was able to amplify *Anisakis* spp. DNA from different sample types, suggesting that the type of fish processing does not affect the quality of the assay by matrix interferences. The specificity and sensitivity of the LAMP method do not seem impaired by sample type, as confirmed by other studies [52]. However, there is a high risk of aerosol contamination due to the large amount of LAMP products. To reduce this risk, we adopted the use of mineral oil inside the reaction tubes.

The LAMP system we described here can detect the concentration of *Anisakis* spp. DNA up to 10^2^ times lower than the RT-PCR method. The high sensitivity of such a method is undoubtedly an advantage, but needs special care to avoid false positives. The utilisation of ready-to-use reagents allowed us to minimise any operator error and the occurrence of possible false positives. Moreover, the presence of non-target DNA and inhibitors in the LAMP reaction was shown to not affect the amplification results. The present method in combination with the ICGENE mini portable instrument proved to be accurate as of the Real-Time PCR method, but more rapid, easy to use, and with a lower limit of detection. Furthermore, the LAMP system proposed in this work is inexpensive, requires little equipment and technical support, is not space-consuming, and is easy to use.

The IC GENE instrument allows one to analyze up to ten samples per test, excluding positive and negative controls, with very low costs. Given the satisfactory results obtained during the validation and the ease of use, the samples do not need to be analyzed in duplicate. *Anisakis* spp. nematodes are considered one of the most critical biological hazards present in seafood products [53].

These food-borne parasites are also a hidden food allergen [54]. Concerning the *A. simplex* complex, there is no information about the minimum allergen concentration that causes allergic diseases. In this respect, [34] concluded that a single larva contains sufficient allergens to induce an antibody response in sensitive individuals. Exposure to this parasite seems to increase due to the increasing consumption of seafood worldwide and the increase of new gastronomic trends based on the consumption of raw and undercooked fish. Because all the *Anisakis* spp. studied originate the same disease and allergic reactions, the species identification is less relevant [24]. To prevent the occurrence of this pathogen in fish sectors, effective and efficient control procedures must be adopted throughout the fish industry and processing continuum. Therefore, it is necessary to use more efficient, simple, and cost- and time-effective methods. Rapid or alternative methods are important, as they reduce the detection time for *Anisakis* spp. considerably in comparison to other inspective methods and prevent the dissemination of these nematodes along the food chain.

The assay validated in this work has the advantages of being more straightforward and more sensitive than PCR and real-time PCR [55]. The LAMP method also has the characteristics of not requiring special reagents and sophisticated temperature control devices so the detection of LAMP products is also suitable for on-site conditions. Furthemore, the preservation of the reagents used for the LAMP assay only requires storage at >4 °C. Therefore, the LAMP assay proposed in this work should be considered a new and reliable tool for food quality and security control for prevention of *Anisakis* allergy in fish sectors.

The present system could be suitable for use by operators of the fish product processing industry for self-monitoring purposes, given its extreme ease and speed, in order to verify the correct evisceration practices of fish sector operators, following the suggestion of the Italian Ministry of Health [22]. Furthermore, due to its capability of discriminating *Anisakis* spp. DNA with respect to species belonging to the *Pseudoterranova* genus, the LAMP system proposed in this work can be a valuable tool to trace Mediterranean fish products.

## 5. Conclusions

The results obtained confirmed the reliability of the method analysed, which is faster and has more sensibility (in terms of limit of detection) than the real-time PCR method compared. In conclusion, the great rapidity, sensitivity, and ease of use suggest that LAMP assay can be a valid alternative for routine examination in the fishery sector to help manufacturers establish concepts for hazard analysis and critical control points (HACCPs) evaluation, where sophisticated and expensive equipment is not sustainable.

## Figures and Tables

**Figure 1 foods-09-00092-f001:**
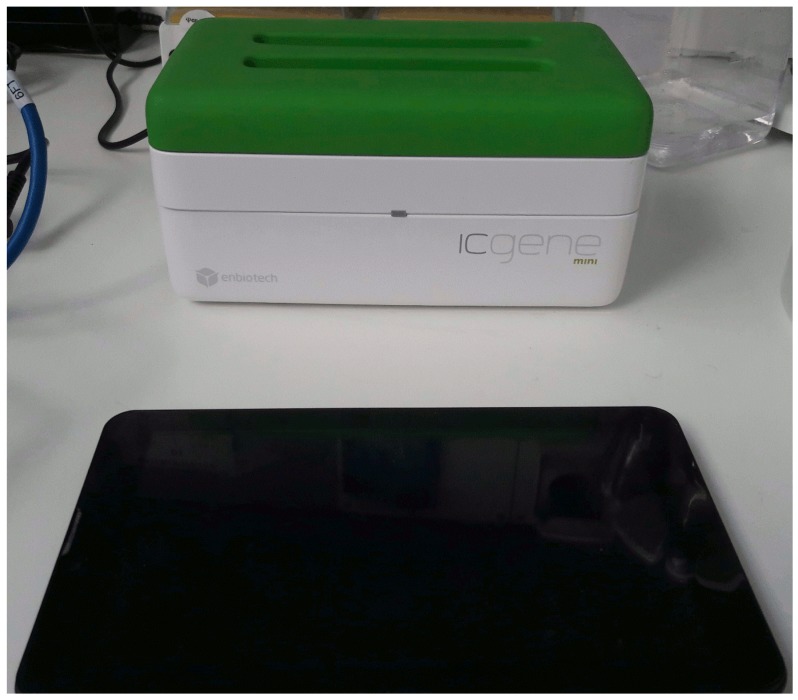
ICGENE mini portable instrument.

**Figure 2 foods-09-00092-f002:**
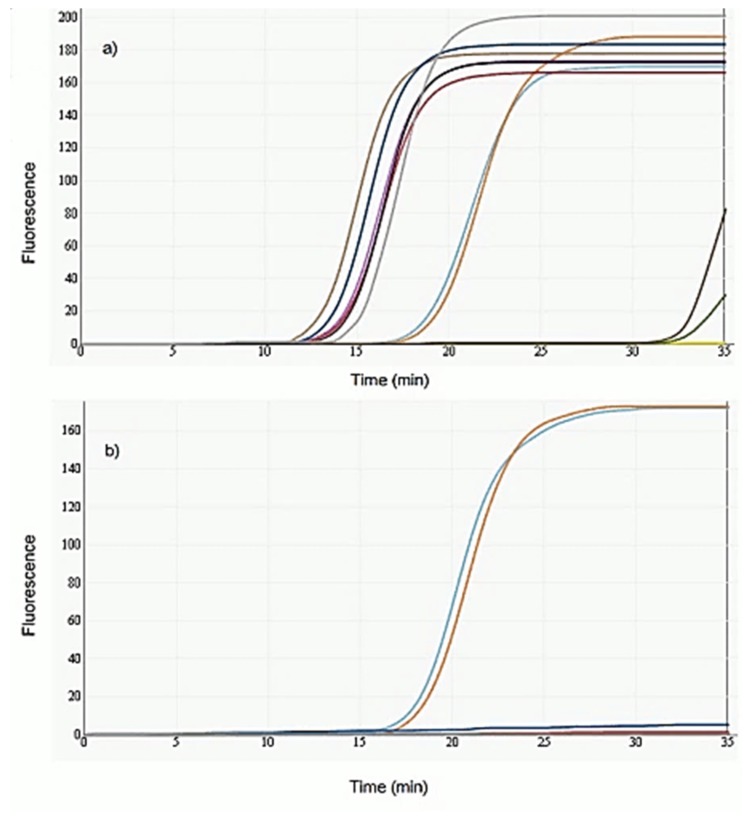
Monitoring of Loop-mediated isothermal amplification (LAMP) amplification for sensitivity (**a**) and specificity (**b**) tests from homogenized farmed salmon samples. The analysis shows the DNA amplification detection in real-time (colored curves shown in (**a**) and (**b**)) by measuring the increasing fluorescence of DNA binding to the dye. The amplification plots displayed in the specificity correspond to the positive control (duplicate).

**Figure 3 foods-09-00092-f003:**
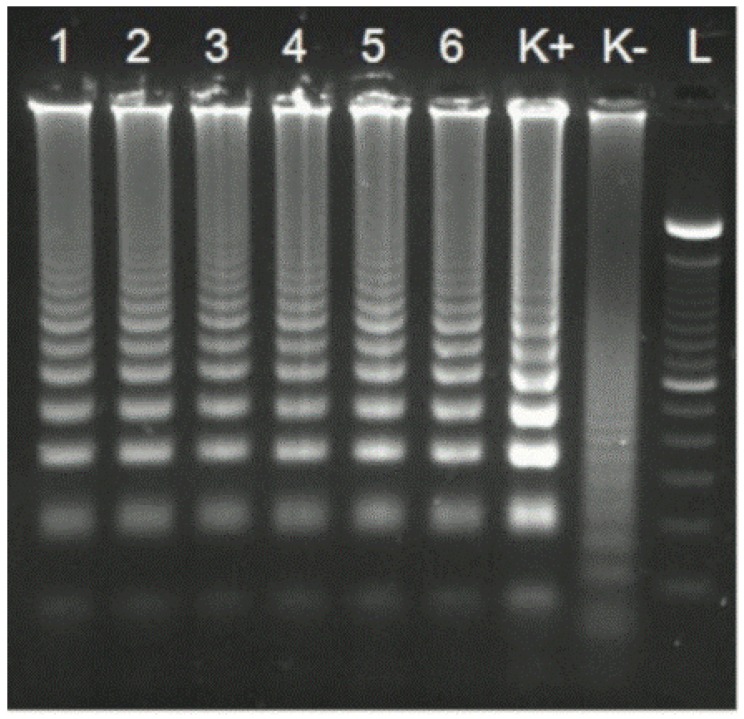
Amplification of LAMP for DNA extracted from anchovy paste samples experimentally infected with *Anisakis* spp. larvae (Lanes 1–6). Lane K+: Positive control, Lane K−: Negative control, Lane L: 100 bp DNA ladder.

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
