# Peer review of "Validation of a Commercial Loop-Mediated Isothermal Amplification (LAMP) Assay for the Rapid Detection of *Anisakis* spp. DNA in Processed Fish Products"

_foods, 2020, doi:10.3390/foods9010092_

Round 1
Reviewer 1 Report
The manuscrip entitled “Validation of a comercial Loop-Mediated Isothermal Amplification (LAMP) assay for the rapid detection of Anisakis spp. DNA in processed fish products” presented by Cammilleri G. et al is a well written paper about a new LAMP assay to detect Anisakis spp. in fish. I have some considerations.
How was the LAMP amplification optimized to performed finally at 65ºC for 35 minutes. By conventional LAMP assay? By real time LAMP assay to follow time of reaction? You should explain this.
Lane 129. Anisakis spp. (spp not in italic)
Lane 157. Anisakis spp. (spp not in italic)
In discussion section (lane 208) the authors say that they adopted the use of amineral oil inside the reaction tubes to reduce the rish of contamination. This possible solution is not indicated in material and methods and it is not clear how the do this kind of assay. Please, indicate in material and methods section.
Authors do not indicate in the manuscript how this LAMP assay can be commercialized. It is necessary to maintain the cold chain to preserve reactives?. Do they need freeze?. How many samples can be analyze in a single test?. I think that only eigh tubes can be used in a single assay, and If at least 2 control samples must be includen in each test (one negative and one positive) just only 3 samples (analyzing in duplicate) can be analyzed. I think this is not profitable for fish products inspection as many fishes must be processed and inspected daily for human consumption. Please, try to discuss this.
And, what about the ICGENE device and application?. You should explain about the device functionality, characteristics, price, etc.
Author Response
Dear Reviewer
Please find attached the revised version of our paper entitled “Validation of a commercial Loop-Mediated Isothermal Amplification (LAMP) assay for the rapid detection of Anisakis spp. DNA in processed fish products”. We have carefully revised our manuscript. We apologize for the delay but we have done our best to respond to the requests of the reviewers in the best possible way, adhering as closely as possible to the high journal's standards. The english grammar of the MS was checked by the software Grammarly. We have attached a word file with the track changes made to ease your perusal of our manuscript changes.
All the step by step changes are reported below:
Reviewer 1
How was the LAMP amplification optimized to performed finally at 65ºC for 35 minutes. By conventional LAMP assay? By real time LAMP assay to follow time of reaction? You should explain this.
Dear reviewer, we confirm that the method was optimised by real time monitoring of the time of reaction. We reported this information on the main document (Lines 160-168).
Lane 129. Anisakis (spp not in italic); Lane 157. Anisakis spp. (spp not in italic)
We have made these changes as suggested
In discussion section (lane 208) the authors say that they adopted the use of a mineral oil inside the reaction tubes to reduce the rish of contamination. This possible solution is not indicated in material and methods and it is not clear how the do this kind of assay. Please, indicate in material and methods section.
Dear reviewer, we apologize for the missing information in the M&M about the use of mineral oil. We provided detailed information of the reaction mix in the main document.
It is necessary to maintain the cold chain to preserve reactives? Do they need freeze?
Dear reviewer, all the reagents need to be stored only at +4°C. We reported this information in the discussion section (Line 270)
How many samples can be analyze in a single test?. I think that only eigh tubes can be used in a single assay, and If at least 2 control samples must be includen in each test (one negative and one positive) just only 3 samples (analyzing in duplicate) can be analyzed.
Dear reviewer, the assay permit to analyse up to 10 samples excluding the negative and positive controls. Given the satisfactory results obtained during the validation of the assay proposed, and the ease of use, we believe that the samples do not need to be analyzed in duplicate. Therefore, we think that 10 samples per test could be enough for daily inspection. We try to discuss this on the main document.
And, what about the ICGENE device and application?. You should explain about the device functionality, characteristics, price, etc.
We reported all the information requested in the main document (Lines 120-130 and lines 254-257).
Hope these changes could be helpful for the manuscript reconsideration.
Thank you for the privilege to consider our work in this esteemed journal.
King regards
Gaetano Cammilleri
Reviewer 2 Report
The problem of human health risks due to fish ingestion is an important issue as foodborne parasitic infections are frequent worldwide due to wild and aquaculture fish. As fish is intended for human consumption, it is important to search for innovative techniques for pathogen’s detection, assuring the food safety and minimizing the zoonotic potential that infected fish pose.
The present article presents the validation of a LAMP assay for the rapid detection of Anisakis spp. in fish. Unfortunately, as it is presented is not accepted. As the paper is based on a technique, it should be reconsidered and resubmitted as a technical report.
Introduction
Very important epidemiological data is missing about the Anisakiasis as zoonosis, not highlighting the importance of the development of this molecular technique.
Information about the life cycle of the parasite and potential hosts, is missing.
A report to the comparative evaluation between routinely applied techniques would underline the innovation of the proposed technique.
Materials and Methods
Information about the fish used as samples are missing. How many fish were used, what was their size, their origin and the sampling season?
How do the authors assure that the samples used were pathogen free?
All the fish samples used were from the same fish or different specimens from the same fish species?
How many samples from each fish species were used in total?
The set of primers designed for the applied technique, should be mentioned in the text.
Results
Detailed results should be presented separately for each fish species.
The results presentation is almost missing.
Discussion
The text is poor and not well-structured.
Comparison to other techniques applied for Anisakis detection should be presented.
Details of the protocol should be discussed.
Author Response
Dear Reviewer
Please find attached the revised version of our paper entitled “Validation of a commercial Loop-Mediated Isothermal Amplification (LAMP) assay for the rapid detection of Anisakis spp. DNA in processed fish products”. We have carefully revised our manuscript. We apologize for the delay but we have done our best to respond to your requests in the best possible way, adhering as closely as possible to the high journal's standards. The English form was checked by the software Grammarly. We have attached a word file with the track changes made to ease your perusal of our manuscript changes.
All the step by step changes are reported below:
The present article presents the validation of a LAMP assay for the rapid detection of Anisakis in fish. Unfortunately, as it is presented is not accepted. As the paper is based on a technique, it should be reconsidered and resubmitted as a technical report.
Dear reviewer, we submitted the work as technical report, according to your precious suggestion.
Very important epidemiological data is missing about the Anisakiasis as zoonosis, not highlighting the importance of the development of this molecular technique.
Dear reviewer, we added more information about the Anisakiasis epidemiology in the main document (Lines 45-50), according to your precious suggestion.
Information about the life cycle of the parasite and potential hosts, is missing.
Dear reviewer, we added more information about the Anisakis life cycle as suggested (see Lines 38-42)
A report to the comparative evaluation between routinely applied techniques would underline the innovation of the proposed technique.
Dear reviewer we added more information regarding the routinely applied techniques and their critical issues in the main document (Lines70-74)
Information about the fish used as samples are missing. How many fish were used, what was their size, their origin and the sampling season?
Dear reviewer, in this work we used only processed fish products from supermarkets and organized large-scale distribution of Italy therefore, it was not possible to trace the size and sampling season of the samples analysed. The homogenized farmed salmon samples came from Norway, whereas all the other processed fish samples used came from Italy. A total of 40 samples were examined for each type of processed fish product. We added this information on the main document
How do the authors assure that the samples used were pathogen free?
Dear reviewer, we considered farmed trout, farmed salmon and farmed sea bream as pathogen free samples based on what is reported in the literature (see Peñalver et al. 2010 https://doi.org/10.4315/0362-028X-73.7.1332; Lunestad et al. 2003 https://doi.org/10.4315/0362-028X-66.1.122; Angot and Brasseur 1993 https://doi.org/10.1016/0044-8486(93)90468-E; EFSA 2010 Scientific Opinion on risk assessment of parasites in fishery products).
All the fish samples used were from the same fish or different specimens from the same fish species?
We used 40 different samples from 5 different fish species, according to what was reported in the modified material and methods section of the main document.
How many samples from each fish species were used in total?
Dear reviewer, we used 40 samples from each processed fish type. All the processed fish types belonged to 5 fish species (Oncorhynchus mykiss, Salmo salar, Sparus aurata, Engraulis encrasicolus and Sardina pilchardus).
The set of primers designed for the applied technique, should be mentioned in the text.
Dear reviewer, unfortunately the set of primers used are subjected to industrial and commercial secrecy restriction, therefore we cannot reveal their sequences. However, we have reported on the main document (Lines 112-118) what was done to obtain the primers set.
Detailed results should be presented separately for each fish species.
Dear reviewer, both the sensitivity and specificity tests carried out in this work showed results equal to 100% (as stated on lines 193 and 202) for all the types of processed fish sample analysed, therefore we believe it could be redundant to report the results obtained for each type of processed fish product.
Comparison to other techniques applied for Anisakis detection should be presented.
Dear reviewer, a comparison with the Real Time PCR protocol proposed by Cavallero et al. 2017 was reported in the main document
Details of the protocol should be discussed.
We added further details of the protocol used in the Materials and Methods section (Lines 120-130).
Finally, we added more information regarding the advantages of the method proposed in order to improve the discussion section, according to your precious suggestions.
Hope these changes could be helpful for the manuscript reconsideration.
Thank you for the privilege to consider our work in this esteemed journal.
King regards
Gaetano Cammilleri
Round 2
Reviewer 1 Report
Thank you very much for answering some of the considerations made above.
After reading the manuscript again I have found some additional consideration:
lane 124: Anisakis spp. (spp not in italic). Please, be carefull
Figure 2: they have to explain what each line is. Please, explain.
Figure 3: how they explain the smear with bands of the line k-. ??
Thank you.
Author Response
Dear Reviewer
Please find attached the revised version of our paper entitled “Validation of a commercial Loop-Mediated Isothermal Amplification (LAMP) assay for the rapid detection of Anisakis spp. DNA in processed fish products”. We have attached a word file with the track changes made to ease your perusal of our manuscript changes.
All the step by step changes are reported below:
lane 124: Anisakis spp. (spp not in italic). Please, be carefull
Dear Reviewer, We have made these changes according to your precious suggestion
2. Figure 2: they have to explain what each line is. Please, explain.
Dear reviewer, we added more information about figure 2 according to your precious suggestion.
3. Figure 3: how they explain the smear with bands of the line k-. ??
Dear Reviewer, the smear present in the negative control samples refers to the reagents used for the LAMP assay. The negative control refers to a product of the Anisakis Screen Glow commercial kit. The negative control did not produce the classical pattern of LAMP amplified product, which is a ladder pattern because LAMP method can form amplified products of various sizes consisting of alternately inverted repeats of the target sequence on the same strand (according to Parida 2008).
Hope these comments could be helpful for the manuscript reconsideration.
Thank you for the privilege to consider our work in this esteemed journal.
King regards
Gaetano Cammilleri
Reviewer 2 Report
The manuscript as it was resubmitted, it was improved in some points but still is not accepted for publication.
Unfortunately, as it is presented, it is not a technical report. It seems more like a validation of the protocol of the commercial kit Anisakis Screen Glow. It should be only a scientific manuscript and not promotion for a commercial kit. It is necessary the set of primers to be presented, in order the paper to be acceptable, as a technical report. It is common the primer sets to be presented, as it raises the acceptability and reference score of the paper from the scientific society.
Discussion needs further improvement. More information should be highlighted about and validity, applications and advantages of commercial kits, especially for fish pathogens.
Author Response
Dear Reviewer
Please find attached the revised version of our paper entitled “Validation of a commercial Loop-Mediated Isothermal Amplification (LAMP) assay for the rapid detection of Anisakis spp. DNA in processed fish products”. We have attached a word file with the track changes made to ease your perusal of our manuscript changes.
All the step by step changes are reported below:
Reviewer 2
The manuscript as it was resubmitted, it was improved in some points but still is not accepted for publication. Unfortunately, as it is presented, it is not a technical report. It seems more like a validation of the protocol of the commercial kit Anisakis Screen Glow. It should be only a scientific manuscript and not promotion for a commercial kit. It is necessary the set of primers to be presented, in order the paper to be acceptable, as a technical report. It is common the primer sets to be presented, as it raises the acceptability and reference score of the paper from the scientific society.
Dear reviewer, we apologize for the inconvenience of not submitting our work as a technical report, as declared during the first round of review; unfortunately, it was not possible to structure and submit it as a technical report due to the fact that the journal does not accept technical reports but only technical notes. However, we would like to point out that we intended to submit the validation of a new commercial kit based on the LAMP method for the research of Anisakis spp. in processed fish products, given the lack of methods in the literature. We also wanted to point out that the literature reports several works concerning the validation of commercial kits based on molecular methods (including the LAMP method) for diagnostic activities in the field of food safety (see https://doi.org/10.1016/j.foodcont.2016.05.035; doi:10.1371/journal.pone.0090545; doi: 10.4315/0362-028X. JFP-14-535; Postel et al. 2010 Evaluation of two commercial loop-mediated isothermal amplification assays for detection of avian influenza H5 and H7 hemagglutinin genes; https://doi.org/10.1016/j.jviromet.2007.08.013; http://dx.doi.org/10.1016/j.jviromet.2012.11.007; https://doi.org/10.1016/j.ijfoodmicro.2017.06.011). Many of them followed and calculated the same validation parameters used in our work. Furthermore, none of the articles mentioned above reported the sequences of the primer set used.
As declared during the first round of revision, we confirm that the set of primers used are subjected to industrial and commercial secrecy restriction, therefore we cannot reveal their sequences. We have reported on the main document (Lines 112-118) what was done to obtain the primers set.
Finally, we also confirm that the article is not intended to be a promotion for a commercial kit. As declared, under my responsibility, in the conflict of interests statement, we avoid entering into agreements with study sponsors, both for-profit and non-profit, that interfere with our access to all of the study’s data or that interfere with our ability to analyze and interpret the data and to prepare and publish manuscripts independently when and where we choose.
We therefore hope for your understanding about these requests.
Discussion needs further improvement. More information should be highlighted about and validity, applications and advantages of commercial kits, especially for fish pathogens.
Dear reviewer, we added more information about the validity and advantages of rapid commercial kits in the discussion section (lines 254-255; lines 267-272; lines 279-282) according to your precious suggestions. Furthermore, we decided to eliminate the cost per reaction as requested by the Reviewer 1 in the first round of revision in order to make it seem as little as possible a promotion of the commercial kit and remaining congruous to what has been declared in other scientific works reported in literature.
Hope these comments could be helpful for the manuscript reconsideration.
Thank you for the privilege to consider our work in this esteemed journal.
King regards
Gaetano Cammilleri